# IS THIS JUST FANTASY?
# LANGUAGE MODEL REPRESENTATIONS REFLECT HUMAN JUDGMENTS OF EVENT PLAUSIBILITY

**Michael A. Lepori**[*]
Department of Computer Science
Brown University

**Jennifer Hu**
Department of Cognitive Science
Johns Hopkins University

**Ishita Dasgupta**
Google DeepMind

**Roma Patel**
Google DeepMind

**Thomas Serre**
Department of Cognitive
& Psychological Sciences
Brown University

**Ellie Pavlick**
Department of Computer Science
Brown University

## ABSTRACT

Language models (LMs) are used for a diverse range of tasks, from question answering to writing fantastical stories. In order to reliably accomplish these tasks, LMs must be able to discern the modal category of a sentence (i.e., whether it describes something that is possible, impossible, completely nonsensical, etc.). However, recent studies have called into question the ability of LMs to categorize sentences according to modality (Michaelov et al., 2025; Kauf et al., 2023). In this work, we identify linear representations that discriminate between modal categories within a variety of LMs, or **modal difference vectors**. Analysis of modal difference vectors reveals that LMs have access to more reliable modal categorization judgments than previously reported. Furthermore, we find that modal difference vectors emerge in a consistent order as models become more competent (i.e., through training steps, layers, and parameter count). Notably, we find that modal difference vectors identified within LM activations can be used to model fine-grained human categorization behavior. This potentially provides a novel view into how human participants distinguish between modal categories, which we explore by correlating projections along modal difference vectors with human participants' ratings of interpretable features. In summary, we derive new insights into LM modal categorization using techniques from mechanistic interpretability, with the potential to inform our understanding of modal categorization in humans.

## 1 INTRODUCTION

Language models (LMs) trained on internet-scale data have demonstrated great success in answering questions about our world (Brown et al., 2020), often displaying a surprising understanding of the physical laws governing it (Gurnee & Tegmark, 2024). However, much of the content on the internet does not accurately reflect the world that we live in — it overrepresents unlikely events (Gordon & Van Durme, 2013), contains innumerable pages of text about fantastical fictional universes, and even contains completely nonsensical sentences (e.g., of the *colorless green ideas sleep furiously* variety (Gulordava et al., 2019), or in the collected lyrics of The Beatles). This raises the question: How do LMs determine whether a sentence describes actual reality, a hypothetical scenario, or something more inconceivable — i.e., how do LMs map linguistic expressions to their corresponding **modal categories**?

---

[*]Corresponding author: michael_lepori@brown.edu

Understanding how LMs represent modal categories is essential for at least two reasons. First, LMs are increasingly deployed as knowledge bases (Petroni et al., 2019), including in high-stakes situations (Magesh et al., 2024; Cheng et al., 2023). Having the ability to distinguish facts about the real world from flights of fancy (and total nonsense!) is a crucial prerequisite for such applications. Secondly, understanding an LM's representation of modal categories can be informative for uncovering the underlying "world model" that it has encoded (Mitchell, 2025; Li et al., 2023). This includes coarse-grained knowledge, such as whether a scenario could happen in *some* possible world, and more fine-grained knowledge, such as whether a scenario is probable, improbable, or impossible in the real world. Modal intuitions have long been used to characterize the folk theories that people employ to understand domains such as physics (McCoy & Ullman, 2019; Shtulman & Carey, 2007; Shtulman & Morgan, 2017). Probing LM representations of the same categories can 1) reveal whether LM representations of modal categories are concordant with human intuitions about these categories, and 2) potentially be used to evaluate whether an LM encodes aspects of the real world that are related to event plausibility.

However, recent work has questioned the ability of LMs to distinguish between modal categories, arguing that their sensitivity to surface-level features makes LM probability estimates a poor indicator of a sentence's modal category (Kauf et al., 2023; Michaelov et al., 2025). This is not unexpected, as a wide variety of considerations, aside from modal category, factor into next-token probability judgments (McCoy et al., 2024). The key question remains: are modal categories represented as coherent features in their own right *within* the LM, or do LMs only represent modality implicitly through unreliable probability estimates?

While Kauf et al. (2023) provides some preliminary indication that models may contain such internal representations using a probing analysis, we substantially elaborate on these results by expanding the range of modal categories and datasets under consideration. We analyze the development of internal representations of modal categories, relate these internal representations to human participants' categorization behavior, and show how we can interpret them in terms of human-understandable features. Specifically, we aim to answer the following research questions:

**RQ 1** *Do LMs have internal representations of modal categories that go above and beyond merely representing output probabilities?*

**RQ 2** *How do LM representations of modal categories develop **a)** over the course of training **b)** over consecutive layers **c)** as model size increases?*

**RQ 3** *Do LM representations of modal categories reflect fine-grained human categorization decisions?*

**RQ 4** *What interpretable features do these representations correspond to?*

To foreshadow our results, we find that LMs often do contain linear representations of the difference between modal categories, or modal difference vectors, which can be used to classify stimuli drawn from a variety of existing datasets. Modal difference vectors are often more discriminative than the probability of the sentence (RQ 1). We find that modal difference vectors emerge in an intuitive order, with more obvious categorical distinctions emerging earlier in training/layers/scale than more fine-grained distinctions (RQ 2). For both evaluations, we rely on expert labels of modal categories. However, human participants' intuitions of modal categories are not bound to reflect such expert labels. Thus, to address RQ 3, we analyze human behavioral categorization data, finding that projections of sentences onto modal difference vectors yield a feature space that reflects human categorization distributions. Intuitively, this feature space clusters stimuli by modal category. Stimuli that lie in between clusters engenders greater disagreement among human participants. Finally, we find that some modal difference vectors correspond to interpretable sets of features, such as subjective event likelihood or imageability (RQ 4).

Overall, our results provide evidence that LMs learn to represent the difference between real life and mere fantasy to a greater extent than implied by previous research. Additionally, these representations appear to capture nuanced aspects of human categorization. These results raise the exciting possibility that one might use an LM's representations of modal categories to gain deeper insight into how and whether they have encoded the causal principles that underlie our world, while retaining the ability to imagine hypothetical realities[1].

---

[1]Code available here.

## 2 BACKGROUND

**Modal Categories**  The modal category of a statement describes whether that statement could, could not, or must be true (Mallozzi et al., 2024). The investigation of these categories has a long history in philosophy, where important arguments hinge on the validity of modal statements (Hume, 1739; Kripke, 1980; Gendler & Hawthorne, 2002). For example, the modal premise that *"philosophical zombies are conceivable"* can yield the modal conclusion that *"it is possible that the mind is distinct from the body"* (Chalmers, 1997; Descartes, 1641). Modal categories have a shorter (though still substantial) history in the cognitive sciences, where researchers have extensively studied the modal intuitions of children and adults (Shtulman & Carey, 2007). By probing their intuitions about the (im)possibility of different scenarios, one can obtain a nuanced picture of a participant's intuitive theories about the causal structure of the world (Griffiths, 2015; Shtulman & Morgan, 2017). For example, participants' intuitions regarding the difficulty of magical spells tend to be proportional to how much that spell violates their intuitive theories of physics (e.g., conjuring a frog out of nothing would be more difficult than teleporting a frog) (McCoy & Ullman, 2019). Following previous computational studies of modality (Kauf et al., 2023; Hu et al., 2025a;b), we study the following modal categories:

> **Probable**: Scenarios that are both possible and commonplace in our world. E.g., *chilling a drink using ice*
>
> **Improbable**: Scenarios that are possible, but not commonplace in our world. E.g., *chilling a drink using snow*
>
> **Impossible**: Scenarios that are not possible in our world, as they violate some known law of nature (e.g., physics, biology, etc.). These scenarios might be true in a counterfactual world with different laws of nature. E.g., *chilling a drink using fire*
>
> **Inconceivable**: Scenarios that cannot be evaluated for possibility in any possible world, due to some fundamental semantic or conceptual error (Hu et al., 2025b). We study inconceivable sentences that arise due to selectional restriction violations — unmet requirements that a verb imposes on its arguments (e.g., animacy, concreteness, etc.) (Chomsky, 1965; Katz & Fodor, 1963). E.g., *chilling a drink using yesterday*

**Related Work**  The present study relates to a burgeoning literature investigating world models in LMs — underlying sets of causal principles that the LM encodes to represent and make inferences about the world (Mitchell, 2025). LMs have shown reasonably strong capabilities in encoding the state of a simplified or abstract world (Nanda et al., 2023; Kim & Schuster, 2023; Li et al., 2025; Ivanitskiy et al., 2023), but have struggled when presented with more complex worlds (Vafa et al., 2024). However, a world model must be far richer than a representation of states — it must represent the principles that explain the dynamics and constraints of the world (Ha & Schmidhuber, 2018; Ivanitskiy et al., 2023; Milliere & Buckner, 2024). This literature directly connects to previous work behaviorally assessing LMs' commonsense reasoning capabilities, which implicitly or explicitly assess an LM's understanding of such basic causal principles (Levesque et al., 2012; Zellers et al., 2019; Bisk et al., 2020; Ivanova et al., 2024).

Similar phenomena are studied in the cognitive sciences, where researchers investigate the world models of children and adults through their *intuitive theories* of physics, psychology, and other domains (Carey, 2000; Spelke & Kinzler, 2007; Ullman, 2015). Rather than being complete and precise representations of physical laws, these theories comprise the basic principles that human beings use to make sense of the world around them. Notably, these intuitive theories are imperfect, leading to a variety of incorrect physical inferences (Ullman et al., 2017). However, they are sufficient for operating in the world under normal circumstances. The causal principles comprising intuitive theories directly determine human intuitions about modal categories: probable and improbable events are consistent with these principles, impossible events are inconsistent with these principles, and inconceivable events might violate the basic conceptual presuppositions underlying these principles (Sosa & Ullman, 2022; Hu et al., 2025b). In this work, we analyze LMs' representations of modal categories to gain insight into the world models they have encoded.

Table 1: Properties of the datasets analyzed in the study. **Prob.**, **Improb.**, **Imposs.**, **Inc.** denote whether the dataset contains probable, improbable, impossible, or inconceivable sentences, respectively. **Pair** denotes that the dataset contains minimal pairs that vary in modal category, allowing for classification using either vectors or probability estimation. **Adv.** indicates that the dataset contains pairs that are adversarial in some way (see main text). **Human** indicates that we analyze human behavioral data from this dataset.

| Name | Prob. | Improb. | Imposs. | Inc. | Pair | Adv. | Human |
|------|-------|---------|---------|------|------|------|-------|
| **Hu et al. (2025b)** | ✓ | ✓ | ✓ | ✓ | ✓ | ✗ | ✓ |
| **Goulding et al. (2024)** | ✓ | ✓ | ✓ | ✗ | ✓ | ✗ | ✓ |
| **Vega-Mendoza et al. (2021)** | ✓ | ✓ | ✗ | ✓ | ✓ | Semantic | ✗ |
| **Kauf et al. (2023)** | ✓ | ✓ | ✗ | ✓ | ✓ | Lexical | ✗ |
| **Hu et al. (2025a)** | ✓ | ✗ | ✗ | ✓ | ✗ | ✗ | ✓ |
| **Tuckute et al. (2024)** | ✗ | ✗ | ✗ | ✗ | ✗ | ✗ | ✓ |

## 3 STUDY 1: LMS LINEARLY REPRESENT MODAL CATEGORIES

In this section, we address RQ 1 by first identifying modal difference vectors — linear representations that distinguish between modal categories — from one dataset, and then assessing whether modal difference vectors can be used to classify stimuli from other datasets. We compare this method to classification based on sentence probability (among other baselines), and find that modal difference vectors enable more reliable modal categorization.

### 3.1 METHODS

**Datasets** We use the Hu et al. (2025b) dataset to identify modal difference vectors, as it contains minimal pairs of stimuli for all pairs of modal categories under consideration. Notably, the impossible stimuli belong to that modal category due to violations of physical laws (e.g., *Someone baked a cake inside a freezer.*). The inconceivable stimuli belong to that category because they violate selectional restrictions based on concreteness (e.g., *Someone baked a cake inside a sigh.*). Additionally, Hu et al. (2025b) estimates that these sentences appear in natural text corpora with approximately equal frequency across the four modal categories, reducing the plausibility that simple frequency-based heuristics can be used to classify sentences. We evaluate the identified modal difference vectors using three other datasets; the Goulding et al. (2024), Vega-Mendoza et al. (2021), and Kauf et al. (2023) datasets. These datasets represent different forms of generalization: the Goulding et al. (2024) dataset contains stimuli that are impossible due to other factors (e.g., biology: *Someone is about to be born with 2 wings.*), whereas the Vega-Mendoza et al. (2021) and Kauf et al. (2023) datasets contain sentences that are inconceivable due to animacy violations (e.g., *The laptop bought the teacher.*). By evaluating modal difference vectors on these datasets, we can understand whether they represent modal categories (such as impossibility) *in general*, or whether they instead capture more fine-grained notions like physical impossibility.

Furthermore, the Vega-Mendoza et al. (2021) dataset contains adversarial sentence pairs, where an inconceivable stimulus contains semantically-related terms and an improbable stimulus contains semantically unrelated terms (e.g., *The scientific research was funded by the {microscope/traveler}.*). Finally, the Kauf et al. (2023) dataset contains lexically-adversarial sentence pairs, where the inconceivable and probable sentences contain the same words in a different order (e.g., *The teacher bought the laptop.*) Every dataset contains expert labels of the modal category of all sentences. See Table 1 for a comparison between the datasets used in this study.

**Models** We study a variety of models across scales and families including GPT2-{Small, Medium, Large, XL}, Llama-3.2-{1B, 3B}, OLMo-2-{1B, 7B, 13B}, and Gemma-2-{2B, 9B}.

**Modal Difference Vectors** We create linear representations of the difference between modal categories using Contrastive Activation Addition (CAA), a technique used to create linear representations of concepts (Panickssery et al., 2023). These representations are simply directions in the hidden state

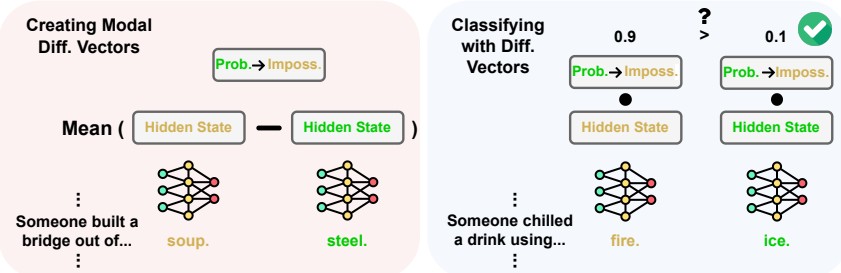

Figure 1: (Left) Diagram describing how we create modal difference vectors. In this example, a modal difference vector capturing the difference between probable and impossible stimuli is created by taking the mean over differences in hidden representations. (Right) Diagram describing how modal difference vectors are used to classify novel minimal pairs of impossible/probable sentences. Hidden representations from each sentence are projected onto the modal difference vector, and the magnitudes of these projections are compared.

space of an LM that correspond to the difference between pairs of stimuli. Because we are concerned with the difference between modal categories, we call these linear representations modal difference vectors.

CAA creates modal difference vectors using pairs of stimuli, $(x_+, x_-)$. $x_+$ expresses one category, and $x_-$ expresses another category. These stimuli are given to an LM, $M$, in separate inference passes, and representations of some token are extracted at a particular layer $l$. $r_+ = M_l(x_+)$, $r_- = M_l(x_-)$. Difference vectors for each pair are computed as $v = r_+ - r_-$. Modal difference vectors are estimated by averaging over many of these single-pair difference vectors. To classify held-out pairs of stimuli $(x'_+, x'_-)$ using a modal difference vector $\bar{v}$, we simply check whether $x'_+ \cdot \bar{v}$ is larger than $x'_- \cdot \bar{v}$ (Marks & Tegmark, 2024). This is analogous to prior work that classifies stimulus pairs based on the overall probability of each sentence (Kauf et al., 2023; Michaelov et al., 2025).

Concretely, we create separate modal difference vectors for every unique pair of categories in {probable, improbable, impossible, inconceivable} by taking the difference between representations of the final "." token at some layer. That layer is found independently for each pair of categories by 5-fold cross-validation, using the classification method described above. If there are ties, we select the median layer that achieved the best performance. After identifying the best layer, we recompute the modal difference vector over all minimal pairs in the Hu et al. (2025b) dataset. See Figure 1 for an illustration of creating modal vectors and classifying stimulus pairs with them.

**Baselines**    We include three baseline classification methods: Probability, Principal Component, and Random. First, we compare to a probability-based classifier. We follow prior work (Kauf et al., 2023) by calculating sentence probability as the sum of the log-probability of each token in the sentence. We expect that $p(inconceivable) < p(impossible) < p(improbable) < p(probable)$ (Hu et al., 2025b). If a minimal pair exhibits this relationship between two stimuli corresponding to different modal categories, then we consider the model to be correct. Next, we compare to a classifier that uses projections along principal components of the hidden states. We compute the first three principal components of the final token of all of the sentences in the WikiText validation corpus, for each layer of each model (Merity et al., 2016). We then run the same cross-validation procedure described above to find the principal component that best partitions stimuli for each pair of modal categories. Finally, we repeat this process with randomly sampled vectors from each layer.

## 3.2    RESULTS

We present classification accuracies on all generalization datasets for all models with at least 2B parameters in Figure 2. We find a qualitative difference in generalization set performance between models that are below this scale, and so we discuss those models separately (see Figure 3, Top Left). This is consistent with Kauf et al. (2023), who also noted that modal categorization ability correlated with scale. For models with at least 2B parameters, we see that for all pairs of modal categories

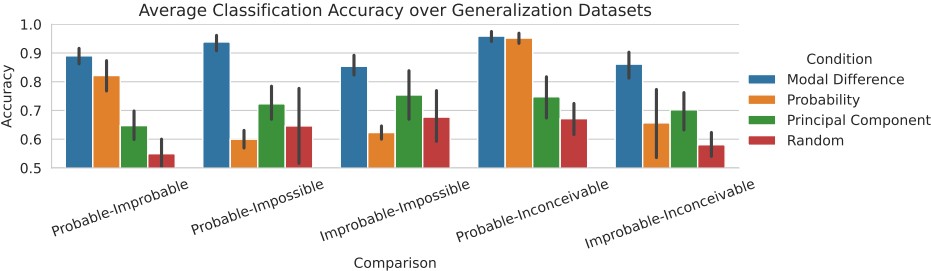

Figure 2: Classification evaluations for models with at least 2B parameters. Results are averages across models and generalization datasets. Modal difference vectors outperform probability estimates and other projection-based classification baselines.

present in the generalization datasets, modal difference vectors match or (sometimes drastically) outperform probability estimates at classifying stimuli by their modal category. Modal difference vectors similarly outperform other projection-based baselines. Notably, this result holds when just considering the adversarial stimuli (See Appendix B).

The results are substantially less clear for models with <2B parameters (See Figure 6, Appendix A). In all cases, modal difference vectors perform worse for these models than they do for larger models. In the case of probable vs. inconceivable distinctions, probability estimates result in higher classification accuracy than modal difference vectors, indicating that there is not one direction that reliably accounts for selectional restriction violations based on both animacy and concreteness. Unless otherwise noted, we will proceed by analyzing only models with at least 2B parameters.

One might worry that these modal difference vectors are epiphenomenal (i.e., not causally implicated in model behavior). To address these concerns, we provide preliminary evidence that one can successfully steer model generations using modal difference vectors in Appendix C.

## 4 STUDY 2: THE DEVELOPMENT OF MODAL CATEGORIES

Prior human studies have found that the ability to distinguish modal categories arises gradually throughout development, with younger children struggling to distinguish between improbable and impossible events (Shtulman & Carey, 2007; Shtulman, 2009). In this section, we address RQ 2 by characterizing how modal difference vectors emerge as a function of various forms of model "development": model scale, layer depth, and training steps. Concordant with prior work (Saxe et al., 2019; Fel et al., 2024), we find that more coarse distinctions emerge in smaller models, in shallower layers, and earlier than more fine-grained distinctions.

### 4.1 METHODS

**Models**   For scaling analyses, we study the complete set of models described in Section 3. For layer depth analysis, we analyze all models with at least 2B parameters. For analyzing training dynamics, we follow Hu et al. (2025b) and study OLMo-2-7B, as this model is accompanied by regular checkpoints throughout training.

**Datasets**   We limit our focus to the Hu et al. (2025b) dataset for this section, as it contains all pairs of modal categories, with no adversarial pairs.

**Analysis**   For analyzing model scale and training dynamics, we run the 5-fold cross-validation pipeline described in Section 3 and report cross-validation performance on the best layer. When analyzing layer depth, we report cross-validation performance for each layer.

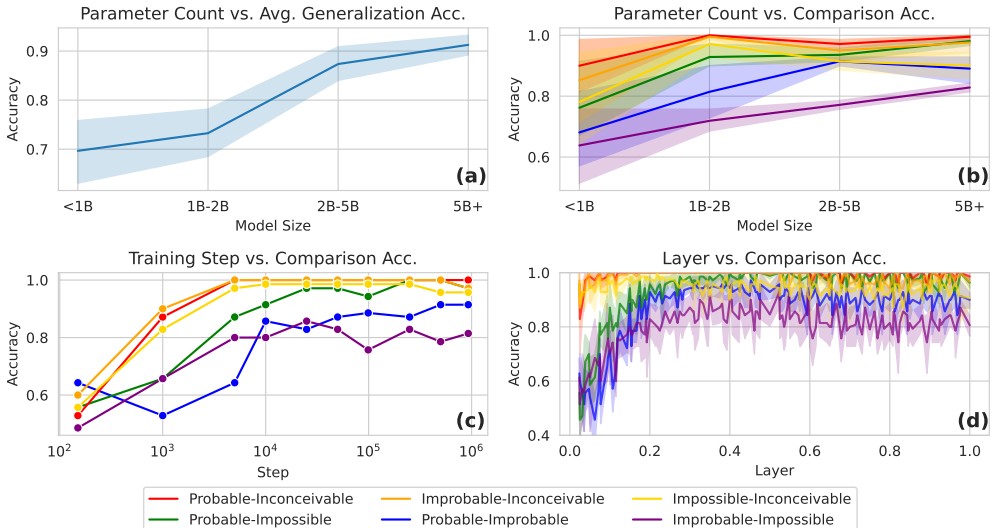

Figure 3: (a) Average generalization performance vs. parameter count reveals a large performance gap between models with fewer/greater than 2B parameters. At (b) smaller scales, (c) earlier in training, and (d) in earlier layers, models form modal difference vectors that can differentiate inconceivable stimuli from other modal categories. After that, models learn the distinction between **probable and impossible**, then **probable and improbable**, and finally **improbable and impossible**.

## 4.2    RESULTS

We present results from model scale, layer depth, and training dynamics analyses in Figure 3. We find that there is a qualitative difference in model performance on generalization datasets between models with <2B parameters and those with ≥2B parameters, as mentioned in Section 3. Aside from that, we find that models first distinguish the inconceivable modal category from the rest, perhaps by taking advantage of distributional signals of selectional restriction violations (Kauf et al., 2023). Models next distinguish probable and impossible events, then probable and improbable, and finally improbable and impossible. Intuitively, we find that more coarse-grained modal distinctions are available earlier than more fine-grained modal distinctions. This pattern of results replicates and expands the results found in Hu et al. (2025b), which analyzes the surprisal assigned to sentences expressing different modal categories over training. We identify the same pattern, except in terms of internal representations. Hu et al. (2025b) also investigates whether the surprisal assigned to sentences expressing different modal categories changes as a function of parameter count. Whereas that work does not identify substantial differences with scale, we find that internal representations develop as a function of parameter count.

## 5    STUDY 3: MODELING HUMAN CATEGORIZATION BEHAVIOR

In all previous analyses, we assumed that the modal category assigned to a stimulus by researchers is the gold-standard label. However, we know that modal categories are graded (Shtulman & Morgan, 2017; Hu et al., 2025a;b), and categorization appears to rely on fuzzy intuitive theories (McCoy & Ullman, 2019). In this study, we address RQ 3 by analyzing whether modal difference vectors can be used to capture human participants' categorization behavior, which does not precisely reflect expert labels.

### 5.1    METHODS

**Datasets**    We use human categorization data from Hu et al. (2025b), Goulding et al. (2024), and Hu et al. (2025a). Hu et al. (2025b) contains categorization data for all four modal categories under study. Goulding et al. (2024) contains data from adult and children participants categorizing probable, improbable, or impossible stimuli as either possible or impossible. We analyze the adult classification

data. Hu et al. (2025a) contains data from adult participants categorizing probable and inconceivable sentences as either "total nonsense" or "not total nonsense". We subset the data to only include stimuli that were classified by four or more participants.

**Analysis** We wish to model human participant's modal categorization intuitions at the stimulus level. To do this, we fit logistic regression models to predict the response distribution of a population of human participants tasked with categorizing a scenario by its modal category (e.g., the proportions of human participants that labeled a scenario as "probable", "improbable", "impossible", and "inconceivable"). To derive a feature space for the logistic regression models, we take all stimuli within a dataset and project them onto three modal difference vectors: probable-improbable, improbable-impossible, and impossible-inconceivable. These vectors are chosen to minimize collinearity between the projections. This defines a 3-dimensional feature space, where stimuli are represented as points within this space. We train logistic regressions to predict the full response distribution for each stimulus using this feature space. Logistic regressions are trained using full batch gradient descent with cross-entropy loss using soft labels. We use the Adam optimizer for 200 epochs Kingma & Ba (2017) and a learning rate of 0.01. We provide a qualitative visualization of this feature space in Figure 4 (Left).

We use leave-one-out cross-validation to predict the response distribution of each scenario. We then compute several metrics to characterize the relationship between predicted and empirical response distributions.

**Baselines** We use the same set of baselines as in Section 3: summed log-probability, projections along principal components, projections along random vectors. We use each of these methods to generate feature spaces on which to fit logistic regression models. Notably, summed log probability only naturally provides a 1-dimensional feature space. However, the other two baselines provide 3-dimensional feature spaces. For each of these feature spaces, we follow the exact same procedure as described above to model the human data.

## 5.2 RESULTS

In Figure 4 (Right), we present several analyses characterizing how the different logistic regression models fit the human data. First, we compute the overall correlation between the empirical and predicted response distributions. Specifically, we report the correlation between the empirical and predicted probabilities assigned to $N - 1$ of the categories for each stimulus, where $N$ is the number of classes (as there are $N - 1$ degrees of freedom in each distribution). This correlation provides a coarse measure of the relationship between probability distributions — a fairly high value might be achieved by merely correctly predicting the response distribution for stimuli that clearly belong to one modal category. Thus, we also characterize the averaged mean squared error between predicted and empirical response distributions. Finally, we correlate the entropy of empirical and predicted response distributions. Across all analyses, we find that a feature space derived from modal difference vectors routinely outperforms the baselines. Additionally, we provide qualitative examples of stimuli and their predicted and empirical response distributions in Table 2. Overall, we find that modal difference vectors reflect participants' graded, intuitive notions of these categories better than the baselines.

We investigate ablations of the modal difference vector feature set in Appendix D, and find that all three features are required to capture the fine-grained categorization behavior elicited by the Hu et al. (2025b) dataset. In contrast, subsets of features are sufficient to capture the coarser categorization behavior elicited by the Hu et al. (2025a) and Goulding et al. (2024) datasets.

## 6 STUDY 4: INTERPRETING LINEAR REPRESENTATIONS

One benefit of identifying model-internal representations of modal categories is that they can be directly analyzed in order to understand which features drive modal categorization in LMs. In this exploratory study, we investigate RQ 4 by correlating projections of sentences along modal difference vectors with human ratings of these same sentences along a variety of interpretable dimensions. We study projections onto the three vectors used in Section 5. Interpreting these vectors might elucidate the relationship between the different modal categories, which is currently an open question (Hu et al., 2025b).

Table 2: Qualitative examples from Goulding et al. (2024) showing the probability estimates that each scenario is possible, as assessed by 1) the logistic regression model using projections on modal difference vectors from Gemma-2-9B, 2) the logistic regression model using probability estimates from Gemma-2-9B, 3) and the proportion of participants classifying the scenario as "possible".

| Scenario (*Someone is about to...*) | Modal Diff. P(Poss.) | Prob. P(Poss.) | Human P(Poss.) |
|---|---|---|---|
| **clean a car.** | 0.99 | 0.70 | 1.0 |
| **clean a road.** | 0.94 | 0.62 | 0.97 |
| **clean a cloud.** | 0.09 | 0.57 | 0.05 |
| **stay awake for 5 hours.** | 0.94 | 0.59 | 1.0 |
| **stay awake for 5 days.** | 0.67 | 0.63 | 0.53 |
| **stay awake for 5 years.** | 0.25 | 0.60 | 0.05 |

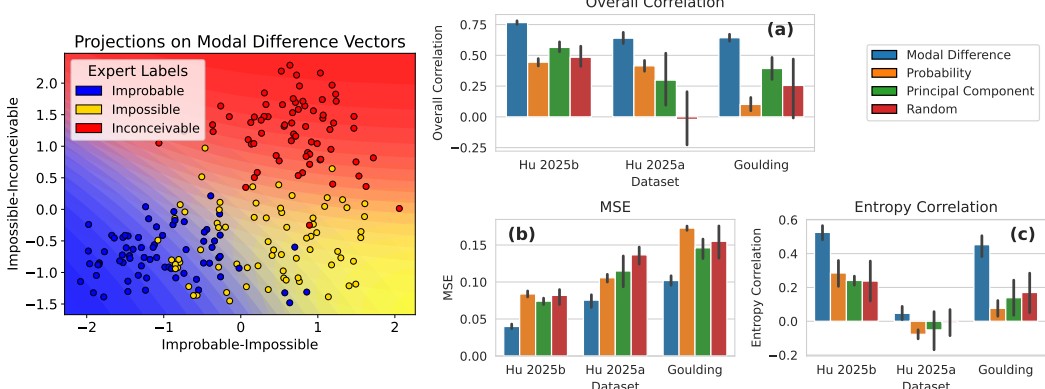

Figure 4: (Left) A qualitative example of stimuli from Hu et al. (2025b) projected along two modal difference vectors. Dots are colored according to their expert label. Background color intensity represents the probability that each point belongs to a particular class according to a logistic regression model fit to this subset of data using these two features. (Right) (a) Pearson correlation between the predicted probability distributions and the empirical proportion of participants that selected each category. (b) Mean squared error between predicted and empirical response distributions. (c) Pearson correlation between the entropy of predicted and empirical response distributions. In all analyses, we find that featurizing using projections along modal difference vectors leads to better models of human categorization behavior.

## 6.1 METHODS

**Datasets** We use human ratings from Hu et al. (2025b), Hu et al. (2025a), and Tuckute et al. (2024). Hu et al. (2025b) contains human participant's ratings of the subjective event likelihood (i.e., "how probable is a scenario?") on a Likert scale for all sentences in the Hu et al. (2025b) dataset (**Event Likelihood** in Figure 5). Hu et al. (2025a) contains 12 sentences from Hu et al. (2025a) annotated according to their average rank in a forced-ranking version of the same subjective event likelihood task (**Ranked Inconceivability** in Figure 5). Additionally, Tuckute et al. (2024) contains 2000 short, diverse sentences, annotated on a Likert scale along a variety of dimensions, including how easy a sentence is to imagine, how grammatical the sentence is and whether the sentence contains a strong emotional valence. This dataset also contains various probability estimates from e.g., another LM or an N-gram model. See Appendix E for descriptions of each dimension.

## 6.2 RESULTS

We project all sentences in each dataset onto the three modal difference vectors used in Section 5: probable-improbable, improbable-impossible, impossible-inconceivable. We then correlate these sentence projections with the human participants' annotations of interpretable features, as discussed above.

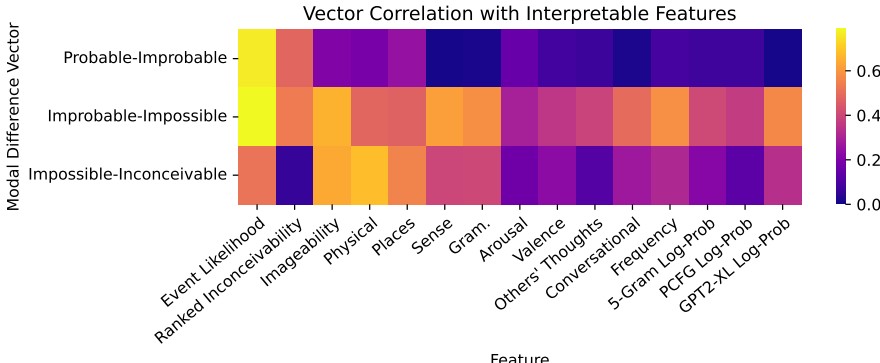

Figure 5: Absolute correlations between projections along modal difference vectors and interpretable features (averaged over models). Notably, Probable-Improbable correlates with human subjective event likelihood judgments, and Impossible-Inconceivable correlates selectively with imageability, the presence of physical objects, and places/environments.

We present the results of this analysis in Figure 5. Reassuringly, we find that projections along the probable-improbable vector correlate very well with subjective event likelihood, but less well with any of the other dimensions. Projections along the improbable-impossible vector are correlated with several features, including subjective event likelihood, imagability, whether the sentence makes sense, and grammaticality. This dispersion of correlations makes it harder to interpret exactly what distinguishes possible from impossible scenarios.

Most interestingly, we see that projections along the impossible-inconceivable vector correlate selectively with features that measure whether a sentence is easy to imagine (either directly or indirectly). This suggests that the ability to imagine a scenario might be a crucial ingredient in distinguishing impossible from inconceivable events. Notably, imagination has been empirically investigated as a factor in distinguishing impossible from possible scenarios (Shtulman & Carey, 2007; Lane et al., 2016; Tipper et al., 2024), but *not* as a mechanism for distinguishing impossible from inconceivable scenarios. However, this finding is consistent with a classic understanding of conceivability from philosophy (Hume, 1739; Yablo, 1993).

## 7 DISCUSSION

We investigate how and whether LMs represent the modal category of a sentence within their hidden states. We find that (1) LMs form representations of modal categories, and that these representations are more diagnostic than output probability distributions (Section 3); (2) modal difference vectors develop at different points over the course of training, layers, and model size (Section 4); (3) modal difference vectors provide a feature space that reflects human categorization behavior (Section 5); and (4) modal difference vectors may reflect human-interpretable features (Section 6). This investigation lays the foundation for a variety of future studies. First, modal difference vectors encoding the difference between, e.g., impossibility and improbability provide a direct means of testing the intuitive theories that LMs derive about the world from raw text input. For example, one might create a controlled dataset of sentences that instantiate different types of physics violations (similar to McCoy & Ullman (2019) or Ivanova et al. (2024)) and use modal difference vectors to check whether LMs represent each of these scenarios as impossible. This investigation could help reveal the physical constraints encoded by the LMs.

Finally, Sections 5 and 6 raise an exciting possibility: one might use modal difference vectors to generate hypotheses about human representations of modal categories. Section 5 establishes a correspondence between modal difference vectors and human categorization behavior, and Section 6 points to a specific, testable hypothesis: that humans distinguish between inconceivable and impossible events on the basis of imagination. Imagination has been shown to significantly impact adults' estimation of event likelihood (Koehler, 1991), and has been investigated as a strategy that adults and children use to distinguish improbable from impossible events (Shtulman & Carey, 2007; Lane et al., 2016; Tipper et al., 2024; Goulding et al., 2022). However, the role of imagination in discerning inconceivable from impossible events remains unknown.

ACKNOWLEDGMENTS

The authors would like to thank the members of the Brown University Language Understanding and Representation Lab and Serre Lab for their valuable feedback on this project. Additionally, we would like to thank Jojo Yang for her detailed comments on the manuscript. Finally, we would like to thank Rachel Goepner, for proofreading the manuscript. ML is supported by the National Science Foundation Graduate Research Fellowship under Grant No. 2439559.

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

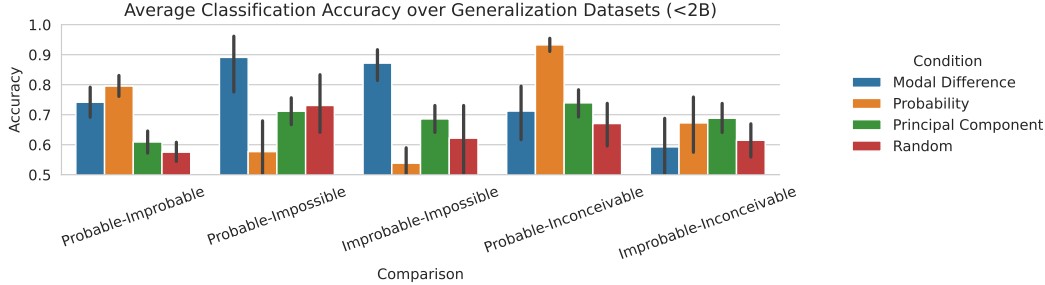

Figure 6: Classification evaluations for models with less than 2B parameters. Results are averages across models and generalization datasets. Results are mixed.

Marten van Schijndel, Andy Exley, and William Schuler. A model of language processing as hierarchic sequential prediction. *Topics in cognitive science*, 5(3):522–540, 2013.

Mariana Vega-Mendoza, Martin J Pickering, and Mante S Nieuwland. Concurrent use of animacy and event-knowledge during comprehension: Evidence from event-related potentials. *Neuropsychologia*, 152:107724, 2021.

Stephen Yablo. Is conceivability a guide to possibility? *Philosophy and Phenomenological Research*, 53(1):1–42, 1993.

Rowan Zellers, Ari Holtzman, Yonatan Bisk, Ali Farhadi, and Yejin Choi. HellaSwag: Can a machine really finish your sentence? In Anna Korhonen, David Traum, and Lluís Màrquez (eds.), *Proceedings of the 57th Annual Meeting of the Association for Computational Linguistics*, pp. 4791–4800, Florence, Italy, July 2019. Association for Computational Linguistics. doi: 10.18653/v1/P19-1472. URL https://aclanthology.org/P19-1472/.

## A  CLASSIFICATION RESULTS FOR MODELS WITH <2B PARAMETERS

We present classification results from models with <2B parameters in Figure 6. We find mixed results across different classification methods, with overall worse performance than with models ≥2B.

## B  CLASSIFICATION RESULTS FOR ADVERSARIAL STIMULI

We highlight the performance of all classification methods on adversarial stimuli. We include lexically adversarial stimuli from Kauf et al. (2023) and semantically adversarial stimuli from Vega-Mendoza et al. (2021). Lexically adversarial stimuli contain sentence pairs with the same tokens, just in a different order (e.g., *The teacher bought the laptop/The laptop bought the teacher*). Semantically adversarial stimuli contain sentence pairs with an improbable stimulus containing a semantically-unrelated word and an impossible stimulus containing a semantically-related word (e.g., *the scientific research was funded by the {traveler/microscope}*. We find that modal difference vectors distinguish all of these cases reliably, whereas other methods distinguish at most one of these types of adversarial stimuli.

## C  STEERING WITH MODAL DIFFERENCE VECTORS

In this section, we demonstrate preliminary evidence that the modal difference vectors can be used to steer the generations of a language model to produce sentences expressing the intended modal category. We assess this using a manually-constructed corpus of 30 novel sentence prefixes. For each prefix and model, we generate continuations as follows: First, we generate the top 5 most likely next tokens. For each of these 5 continuations, we greedily decode 4 more tokens. However, we find that models sometimes generate fragments of run-on or syntactically complex sentences. To generate clean qualitative examples, we store the overall probability of the period token "." after each

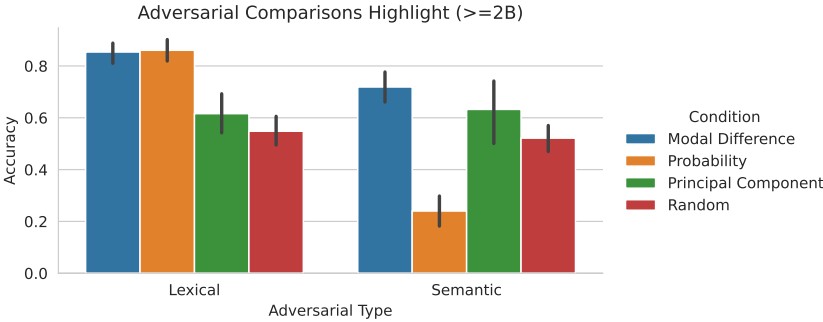

Figure 7: Performance of all classification methods just on adversarial stimuli. We include lexical adversarial stimuli from Kauf et al. (2023) on the left. These stimuli contain the same tokens, just in different orders. We find that modal difference vectors and probability estimates perform best in the face of this manipulation. We include semantically adversarial stimuli used in Vega-Mendoza et al. (2021) and Michaelov et al. (2025) on the right. We find that modal difference vectors and projections along principal components perform best, while probability estimates are systematically misled.

generation. We truncate generations after the token position where the period token received the highest probability.

Following prior work, we intervene on models using modal difference vectors while generating (Panickssery et al., 2023; Ghandeharioun et al., 2024). To do so, we add a scaled version of the modal difference vectors for probable-improbable, probable-impossible, or probable-inconceivable to all residual stream positions at the appropriate layer while generating the next token. We experiment with scalar multipliers of 3 and 5, and find that 5 qualitatively produces better results. As a baseline, we repeat this process with no intervention.

We attempt to steer Gemma-2-2B and Llama-3.2-3B. We present a quantitative analysis of steering success in Figure 8. Here, we use the baseline model (without intervention) to measure the surprisal (or negative log-probability) of the first 5 generated tokens either with or without steering. We find that, for both models, surprisal increases in order of `baseline < improbable < impossible < inconceivable`. This result mirrors the finding from Hu et al. (2025b), indicating that steering has the desired impact on model generations.

We also present several examples from each model on a diverse range of prefixes in Tables 3 and 4. While not perfect, these examples show many instances of steering having the desired effect, rendering generations more improbable, impossible, or inconceivable.

## D STUDY 3 FEATURE ABLATIONS

In this section, we systematically ablate modal difference features from the linear regression models in Section 5 in order to understand whether each feature is necessary for capturing categorization behavior in human participants (See Figure 9). We find that ablating any of the modal difference features significantly harms the overall correlation and MSE metrics for the Hu et al. (2025b) dataset (Bonferroni corrected $p < .05$), which requires the most fine-grained modal categorization judgments. We find that the entropy correlation metric is most sensitive to the Probable-Improbable modal difference feature. We do not observe significant effects of ablating a single feature from the Hu et al. (2025a) and Goulding et al. (2024) datasets, indicating that subsets of the modal difference features are sufficient for capturing coarse-grained (i.e., binary) modal categorization judgments. Significance is computed using a paired t-test comparing each ablation condition to the full-feature condition, with Bonferroni correction applied within each dataset.

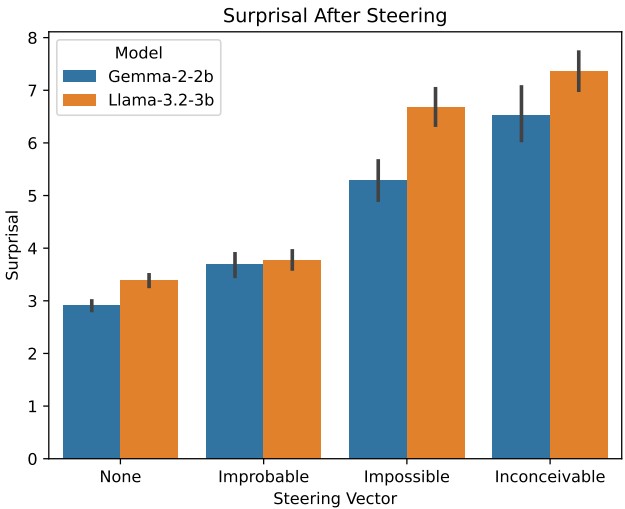

Figure 8: Surprisal (with respect to the baseline model) of the next token predictions generated after steering. We find that, for both Gemma-2-2B and Llama-3.2-3B, surprisal values generally increase in order of `baseline < improbable < impossible < inconceivable`. This mirrors the finding from Hu et al. (2025b).

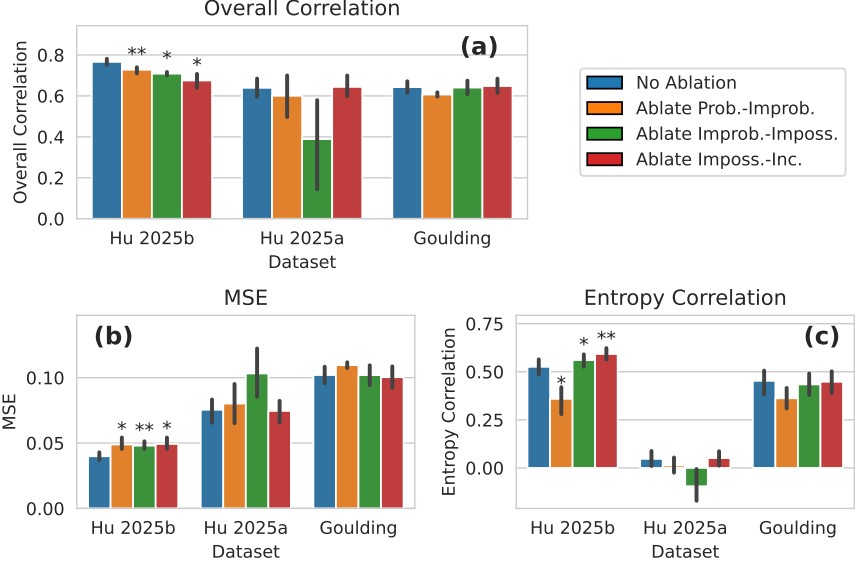

Figure 9: Effect of ablating modal difference features on capturing human participant categorization behavior. For the Hu et al. (2025b) dataset, we find that all three modal difference features are required to best capture behavior (according to Overall Correlation and MSE metrics). In order to capture entropy in the response distribution, regression models require the probable-improbable feature. The impact of ablation is not significant for the other two datasets, which only elicit coarse, binary modal categorization judgments.

Table 3: Steering Generations using Gemma-2-2B.

| Model | Prefix |
|---|---|
| Gemma-2-2B | Someone measured the furniture using a... |
| **Steering** | **Generations** |
| None | tape measure, ruler, 1, measuring tape, laser level |
| Improbable | tape measure, ruler, laser, measuring tape, 3D scanner |
| Impossible | laser, telescope, microscope, new technique, camera |
| Inconceivable | ¡b¿¡i¿¡u¿, team, different scale, word, new one |
| **Model** | **Prefix** |
| Gemma-2-2B | Someone fed the child with a... |
| **Steering** | **Generations** |
| None | spoon, bottle of milk, knife, syringe, glass of water |
| Improbable | spoon, ¡strong¿¡em¿pizza, bowl of soup, baby bottle, pizza |
| Impossible | new song, baby, song, video game, snake |
| Inconceivable | smile, child, good thing, team, different kind of child |
| **Model** | **Prefix** |
| Gemma-2-2B | Someone fixed the car with a... |
| **Steering** | **Generations** |
| None | hammer and a screwdriver, new engine, lot of effort, 1, screwdriver |
| Improbable | hammer, 1, piece of wood, rubber band, sledgehammer |
| Impossible | new car, laser, 3D printer, car, broken mirror |
| Inconceivable | team, smile, good team, ¡em¿team¡/em¿, great team |
| **Model** | **Prefix** |
| Gemma-2-2B | Someone protected the garden with a... |
| **Steering** | **Generations** |
| None | fence, net, protective fence, plastic sheet, wire fence |
| Improbable | fence, 1, giant inflatable, blanket, net |
| Impossible | new song, cloud, virus, song, flower |
| Inconceivable | team, smile, family, new one, ¡em¿team¡/em¿ |

# E  DESCRIPTIONS OF THE FEATURES IN THE TUCKUTE ET AL. (2024) DATASET

In this section, we briefly describe the features in the Tuckute et al. (2024) dataset, which we correlate with projections along modal difference vectors in Section 6.

**Imageability:** Ratings on a 7 point Likert scale, answering the question "how easy is the sentence to visualize, or to form an image of the sentence's meaning in your mind?"

**Physical:** Ratings on a 7 point Likert scale, answering the question "how does the sentence make you think of physical objects and/or physical causal interactions?"

**Places:** Ratings on a 7 point Likert scale, answering the question "how much does the sentence make you think of places, natural scenes, and/or environments?"

**Sense:** Ratings on a 7 point Likert scale, where 1 corresponds to "doesn't make any sense" and 7 corresponds to "makes perfect sense".

**Gram.:** Ratings on a 7 point Likert scale, where 1 corresponds to "completely ungrammatical" and 7 corresponds to "perfectly grammatical".

**Arousal:** Ratings on a 7 point Likert scale, answering the question "how much does the sentence make you feel stimulated, excited, frenzied, wide-awake, and/or aroused?"

Table 4: Steering generations using Llama-3.2-3B

| Model | Prefix |
|---|---|
| Llama-3.2-3B | Someone destroyed a building using a... |
| **Steering** | **Generations** |
| None | sledgehammer, bulldozer, wrecking ball, crane, drone |
| Improbable | flamethrower, bulldozer, sledgehammer, drone, chainsaw |
| Impossible | boat, car, 9, real-life, man and a woman |
| Inconceivable | term, thesium, weapon, gun, a |
| **Model** | **Prefix** |
| Llama-3.2-3B | Someone hit the ball with a... |
| **Steering** | **Generations** |
| None | bat, golf club, racquet, racket, stick |
| Improbable | golf club, bat, baseball bat, racquet, club |
| Impossible | stick, real, boat, golf course, bow |
| Inconceivable | hammer, bat, club, baseball, ball |
| **Model** | **Prefix** |
| Llama-3.2-3B | Someone beat a drum with a... |
| **Steering** | **Generations** |
| None | stick, drumstick, spoon, big stick, hammer |
| Improbable | stick, drumstick, spoon, sword, can of beer |
| Impossible | drum, man in a boat, French, cat, real drum |
| Inconceivable | hammer, heart, mind, head, brain |
| **Model** | **Prefix** |
| Llama-3.2-3B | Someone treated the wound using a... |
| **Steering** | **Generations** |
| None | topical antibiotic, solution of 0, technique, mixture of honey, tourniquet |
| Improbable | home remedy, mixture of honey, loe vera gel, local anesthetic, homemade remedy |
| Impossible | 3, Japanese, bottle, French, real-life |
| Inconceivable | number, series, few, smile, The |

**Valence:** Ratings on a 7 point Likert scale, answering the question "how much does the sentence make you feel happy, pleased, content, and/or hopeful?"

**Others' Thoughts:** Ratings on a 7 point Likert scale, answering the question "How much does the sentence make you think of other people's experiences, thoughts, beliefs, desires, and/or emotions?"

**Conversational:** Ratings on a 7 point Likert scale, answering the question "how likely do you think the sentence is to occur in a conversation between people?"

**Frequency:** Ratings on a 7 point Likert scale, answering the question "how likely do you think you are to encounter this sentence?"

**5-Gram Log-Prob:** Probability estimates from a 5-gram language model.

**PCFG Log-Prob:** Probability estimates from the PCFG parser from van Schijndel et al. (2013).

**GPT2-XL Log-Prob:** Probability estimates from GPT2-XL.

