# OpenReview forum: "Is This Just Fantasy? Language Model Representations Reflect Human Judgments of Event Plausibility"
_ICLR.cc/2026/Conference — ICLR 2026 Poster_

### Official Review · Reviewer_bMoA · 2025-10-29

**Soundness:** 3
**Presentation:** 3
**Contribution:** 3
**Rating:** 6
**Confidence:** 3

**Summary:**

This paper investigates whether LLMs encode human-like distinctions between possible, impossible, and inconceivable events. Using “modal difference vectors” derived from contrastive pairs (e.g., Someone chilled a drink using ice vs. using fire), the authors claim that these linear directions in activation space reflect graded human judgments of plausibility. The paper reports four main studies analyzing representational structure, training dynamics, correspondence to human categorization, and interpretability. Results show that linear separability of modal categories emerges during training and correlates with human judgments. The work contributes to the growing line of research linking mechanistic interpretability with cognitive semantics.

**Strengths:**

This paper's primary strength lies in its originality of problem formulation and the clarity of its investigation. The authors compellingly shift the debate on world models in LM: instead of focusing on output probabilities as an indicator of modal understanding (which prior work has shown to be unreliable), this paper poses a more original and nuanced question about how modal categories are encoded in the model's internal representations. The introduction of "modal difference vectors" as a specific, linear, and geometrically interpretable tool is a novel methodological contribution. The entire paper is well-structured around four precise research questions, which creates a clear, logical narrative that guides the reader from the vector's existence (Study 1) to its development (Study 2), its link to human cognition (Study 3), and its interpretability (Study 4).

The paper's contribution (Study 3) is the demonstration that these modal difference vectors, extracted from LMs, can model fine-grained human categorization behavior better than strong baselines. This moves the field beyond simply asking if LMs "know" the difference between possible and impossible, and instead investigates whether their internal representational geometry captures the graded and fuzzy nature of human modal judgments. This finding is significant because it provides a novel, quantitative framework for comparing the "intuitive theories" of LMs against those of humans.

Finally, the quality of the exploratory analysis in Study 4 is a key strength. The attempt to find interpretable correlates for these abstract vectors is valuable. By correlating vector projections with human ratings of features like "subjective event likelihood" and "imageability," the paper provides a crucial first step toward grounding these internal representations in human-understandable concepts.

**Weaknesses:**

1. There appears to be an inconsistency between the paper's high-level claims and its detailed experimental findings. The abstract makes a general claim that "LMs have access to more reliable modal categorization judgments...". This suggests a general property of the language models studied. However, the paper's own results in Section 3.2 (and detailed in Figure 6, Appendix B ) show that this finding does not hold for models with fewer than 2B parameters. For this group of models, the results are described as "substantially less clear" and "mixed", with the probability-based baseline "sometimes result in higher classification accuracy".

2. The experiment presented in Appendix D, while intriguing, is too small in scale to support a conclusion about the causal properties of the modal difference vectors. The authors state this is "preliminary evidence" and assess it using only "a manually-constructed corpus of 30 novel sentence prefixes" on just two models. This sample size is insufficient to demonstrate a generalizable or reliable phenomenon, providing only anecdotal evidence.

3. A minor point about presentation. The paper's core conceptual framework relies on the distinction between four modal categories (Probable, Improbable, Impossible, Inconceivable). These distinctions, particularly between "Impossible" and "Inconceivable," are subtle and central to the paper's thesis. While the text definitions are provided, a reader not already familiar with this specific literature may struggle to immediately grasp the precise boundaries. It could be improved by including an illustrative figure in the Introduction or Background.

**Questions:**

1.  There appears to be a notable discrepancy between the main claim in the Abstract, which suggests that "LMs" as a general category possess more reliable modal judgments, and the specific results in Section 3.2. Your analysis explicitly states that for models with <2B parameters, the results are "substantially less clear" and that the probability baseline can even "sometimes result in higher classification accuracy". This suggests the paper's core finding is an emergent property of scale. Could you clarify whether you see this as a general property of LMs or a scale-dependent one?

2. Could you provide a bit more rationale on the selection of these specific datasets (e.g., Hu et al. 2025b, Kauf et al. 2023, Goulding et al. 2024)? It would be good to know how this particular combination of datasets helps ensure that the model is learning the abstract concept of modality, rather than potentially learning statistical heuristics that might be common across these specific benchmarks.

3. The steering experiment in Appendix D is an interesting proof-of-concept for the causal effect of the vectors. However, the experiment's scale is very limited, based on "a manually-constructed corpus of 30 novel sentence prefixes" and only two models. While you correctly label this as "preliminary evidence", it is difficult to draw a strong conclusion from such a small sample.

---

> ### Author Response · Authors · 2025-11-20
>
> Thank you for your thoughtful and comprehensive feedback! Find specific responses below.
>
> W1: Thank you for pointing this out! We are happy to be a bit more specific in key sections, like the intro and abstract, regarding the applicability of our findings, and the crucial importance of scale. We will update these sections for the final manuscript submission.
>
> W2: We completely agree, and that is exactly why we chose to put these results in the appendix. Our primary scientific goal in this article is to investigate whether LMs represent modal categories in their activations (in a manner that accords with human judgments of modality), rather than leveraging these representations to control model behavior. This is a key step in advancing the debate over world knowledge in LMs. However, we are interested in using these insights and techniques for more practical applications in the future, such as belief insertion.
>
> W3: Thank you for the suggestion! If space permits, we will try to create such a figure. If not, we will attempt to help the reader by providing more reminders regarding these definitions for the reader throughout the manuscript.
>
> Q1: Great question! I do see this as an effect of scale, though the exact mechanisms by which scale bolsters world knowledge are somewhat opaque at this point. However, we do have suggestive evidence to explain some of the effect of scale. I’ve pasted one of my comments to reviewer 83R5 below, which describes one particular effect of scale that we identify. As mentioned there, we are happy to include some of these interpretations in the Appendix of the final manuscript.
>
> ```
> Figure 6 is simply showing that the modal difference vectors don’t provide extra predictive power over an arbitrary direction in vector space, for models <2B and for comparisons with the inconceivable modal category. In other words, the modal vectors found using the Hu et al. 2025b dataset do not generalize well. My explanation for this is that the model does not derive one consistent direction corresponding to “selectional restriction violations” at this scale. As described above, to achieve good performance on e.g., the Kauf dataset using vectors from the Hu et al. 2025b, the model must internally represent very different selectional restriction violations (on concreteness vs. animacy) consistently within its activations. If these two violations are represented differently, then there is no reason that the modal difference vector would outperform an arbitrary vector.
> ```
>
>
> Q2: These datasets were chosen 1) because they were used in prior literature assessing either LM or human conceptions of modal categories and 2) because they contain diverse sets of stimuli. Section 3.1 contains a description of some of the variation between datasets, but to mention it again here: Hu et al. 2025b contains past tense sentences that are impossible due to physics violation and inconceivable due to concreteness violations. This dataset is used to derive modal difference vectors. However, the generalization datasets also contain impossible sentences due to e.g., biological factors, inconceivable sentences due to animacy violations, future tense sentences, and sentences that are explicitly designed to be adversarial due to either lexical or semantic properties. The commonality between these datasets is that they contain short sentences without surrounding context, but otherwise they are quite diverse along the relevant axes.
>
> Q3: We agree! We look forward to investigating the practical implications of this finding for model control and editing in future work.

---

> > ### Comment · Reviewer_bMoA · 2025-11-21
> >
> > Thank the authors for their clarifications, which have addressed most of my concerns regarding the dataset and the paper's methodology. However, I argue that the intermediate mental states derived through reasoning chains in the method are not entirely reliable—numerous existing works have highlighted this issue (e.g., Barez, Fazl, et al. "Chain-of-thought is not explainability." Preprint, alphaXiv (2025): v1). Strengthening the reliability of this component would further enhance the method.

---

> > > ### Author Response · Authors · 2025-11-21
> > >
> > > Thank you for your reply. Our work does not have anything to do with chain of thought/reasoning models, and our methods explicitly relies on analyzing internal activations (not token generations). Perhaps this comment was posted on the wrong openreview page?

---

### Official Review · Reviewer_swG3 · 2025-10-29

**Soundness:** 3
**Presentation:** 2
**Contribution:** 3
**Rating:** 6
**Confidence:** 4

**Summary:**

The paper investigates whether language models (LMs) encode modal categories—probable, improbable, impossible, and inconceivable—via linear structure in their hidden states. Using Contrastive Activation Addition, it derives modal difference vectors from minimal pairs and shows that, for models ≥2B parameters, these vectors classify modality across multiple datasets (including adversarial ones) and typically outperform sentence-probability baselines. Developmentally, coarser distinctions (e.g., inconceivable vs. the rest) emerge earlier in training, shallower layers, and smaller models, with finer distinctions appearing later and at larger scales. Projecting sentences onto three modal vectors yields a low-dimensional feature space that predicts graded human categorization distributions via logistic regression. Correlation analyses suggest the probable–improbable vector tracks subjective event likelihood, while the impossible–inconceivable vector aligns with imageability/physical concreteness. Preliminary steering results hint these vectors can nudge model generations toward targeted modal categories.

**Strengths:**

1) It shows simple linear method could transfer across datasets and model families, beating stronger-than-naïve baselines on modality classification.
2) Uses projections as features to predict human response distributions (not just hard labels), capturing graded intuitions.

**Weaknesses:**

1) Probability is a weak straw baseline for modality; missing stronger comparators (e.g., calibrated logit regressors over tokenwise surprisals, fine-tuned linear probes with controls) and clearer uncertainty estimates/CIs across splits and seeds.

**Questions:**

Can you demonstrate that injecting/removing these vectors causally mediates modality judgments?

---

> ### Author Response · Authors · 2025-11-20
>
> Thank you for your thoughtful review! Find specific responses below:
>
> W1. The present work is in direct conversation with prior work [1, 2], which all uses LM probability judgments to establish that language models struggle to distinguish modal categories. Thus, our primary scientific contribution is that language model activations do separate modal categories, even though these distinctions are somewhat obscured by LM probability estimates. We welcome alternative methods for deriving these distinctions, including methods that outperform CAA. For example, we expect linear probes to perform quite well at finding these directions. This might be a promising avenue for future research, especially work that attempts to practically use such vectors for e.g., unlearning, hallucination detection, belief shaping, etc. However, due to the size of the datasets we are using, and our research goals, we use the simplest method on offer: CAA. Additionally, we include other baselines to demonstrate that the stimuli are not separable using any arbitrary direction in activation space.
>
> Q: Prior work has demonstrated that models struggle to perform such metalinguistic judgments [3], and so we run a generation-steering analysis (Figure 8). This analysis reveals that these vectors can be used to steer model generations in a manner that replicates the surprisal trends identified in prior work [4].
>
> [1] Kauf et al. 2023 Event Knowledge in Large Language Models: The Gap Between the Impossible and the Unlikely
> [2] Michaelov et al. 2025 Not Quite Sherlock Holmes: Pretrained Language Models Cannot Reliably Differentiate Impossible from Improbable Events
> [3] Hu & Frank 2024 Auxiliary Task Demands Mask the Capabilities of Smaller Language Models
> [4] Hu et al. 2025 Shades of Zero: Distinguishing Impossibility from Inconceivability

---

### Official Review · Reviewer_83R5 · 2025-11-01

**Soundness:** 3
**Presentation:** 3
**Contribution:** 3
**Rating:** 6
**Confidence:** 4

**Summary:**

This paper investigates the ability for language models to categorize the modal categories of sentences, i.e. whether an event is possible, impossible, etc. They identify linear representations that separate modal categories, which they call modal difference vectors. They find that modal difference vectors emerge in a consistent order as the number of training steps, parameter count, and layers increase. They also validate these modal difference vectors with human judgments.

In the first study, they focus on four modal categories, probable, improbable, impossible and inconceivable events and test modal difference vectors vs. a logprob based, PCA based, and random baselines and find that in models above 2B, the modal difference vectors perform better at categorizing the modal category of a sentence, but this is not so for models below 2B parameters. In the second study, they examine accuracy on the classification task for models with different parameter counts, modal category distinctions and throughout the training of a sample model (olmo2 7B). They find that more coarse-grained modal distinctions tend to arise earlier and are more accessible for small models. Lastly, they interpret modal difference vectors by correlating projections with human-rated features.

**Strengths:**

**S1**: I appreciate the thoroughness in terms of types of experiments done, as the modal category vectors are analyzed in multiple different settings, from classification accuracy to emergence in scaling to correlation with human judgements. This makes me more confident that the vectors found do correspond to the modal categories, though I still have some doubts (see weaknesses)

**S2**: The paper is overall clear and well written, with the motivation and importance of the problem being addressed up front and each section. I think the results of the paper would be interesting to the community.

**S3**: The topic is interesting to the community and timely, I think the paper would be interesting to those working on hallucination, factuality or related topics.

**Weaknesses:**

**W1**: This paper falls into a category which I call "vector for X". I want to say here that I don't necessarily believe that "vector for X" type papers are bad, but I think that to be sound work, I would want to see some causal analysis or intervention applied to the vectors found. Keeping in mind that linear separability between groups of points in high dimensions is the default, this makes me automatically a bit wary of papers that have this as a main contribution. In order to validate that the vectors found are meaningful, I'd want to check that they influence the model behaviour in some way. Appendix D is a good start but I would want to see a more thorough investigation:

- necessity of the direction: for instance by nulling out the modal subspace and checking that performance drops on a modal classification task

- behavioural linkage: it would be nice to show that steering vectors have an effect on some kind of classification task rather than just through surprisal. For example, can steering actually change model decisions on modal classification tasks(e.g. with a fine-tuned head?)

- comparison to alternatives: how do CAA-derived vectors compare to other methods?

**W2**: It seems like there’s the possibility for some conflation with some other linguistic concepts such as negation, sentiment or other concepts. Could you comment on whether this is a concern or if you attempted to control for this? Specifically, for the inconceivable/impossible sentences, is it possible that they have more negation-like linguistic terms or that they are less grammatical? A further thing that I would like for papers like this is a frequency control (as much as possible) -- the inconceivable and impossible sentences probably appear far less in pretraining data compared to the other categories?

**Questions:**

- The paper identifies a stark qualitative difference at 2B parameters, with even the random baseline outperforming modal vectors, but this finding receives minimal analysis. Can you comment more on this and why you believe this to be the case?

---

> ### Author Response · Authors · 2025-11-20
>
> Thank you for your kind words and thoughtful feedback. Please find specific responses below.
>
> W1: Very interesting suggestions! At a high level, I completely agree with the critique of “vector for x” contributions.This is precisely why nearly all of our main results are about the generalization of the vectors we found on one dataset to other datasets, which vary in myriad ways (e.g., tense, type of impossibility/inconceivability, semantic or lexical adversarial examples, etc.). The primary scientific contribution is not that there exists linear separability between these concepts in one particular dataset, but that these concepts appear to be reasonably separable in general. For this reason, we used the relatively simple and data efficient CAA procedure to generate these directions, and do not attempt to comprehensively survey different methods for creating such vectors.
>
> Nonetheless, we take your broader point that it would be helpful to the community to connect these findings more tightly to various aspects of model behavior. We agree, and are excited about future directions that leverage these modal difference vectors to predict the ability to insert new facts into specific models (combining the present work with, e.g., recent findings about belief editing [1]), among other things!
>
> W2: Thank you for raising these points. We will include a discussion of them in the main text of the final manuscript.
>
> In general, you are correct that there is room for conflating modality with other linguistic properties. However, we leverage several controlled datasets from prior work in the cognitive sciences, and so we do not find this to be a substantial issue for the present study. Upon inspection, one will not identify correlation between modality and e.g., sentiment or negation.
>
> The most potentially troublesome modal category is likely inconceivability, where sentences like “Someone baked a cake inside a sigh.” might be construed as less grammatical than either impossible, improbable, or probable sentences. This sentence is inconceivable due to selectional restriction violations based on concreteness, and we use sentences of this type to create our modal difference vectors. However, we assess these modal difference vectors on sentences such as “The laptop bought the teacher”, which exhibit a selectional restriction violation based on animacy. Thus, our findings demonstrate that sentences that exhibit selectional restriction violations (and are thus inconceivable) are separable from sentences that do not, which is our goal.
>
> With respect to frequency concerns, the prior work that introduced the datasets used to generate the modal difference vectors accounted for frequency effects [2]. In Figure 3B of that work, they find that inconceivable sentences are actually slightly more common in a large corpus than the sentences from the other modal categories. The other categories are close to one another in frequency. Thus, we can infer that modal difference vectors should not be largely confounded by frequency.
>
> Furthermore, our modal difference vectors empirically outperform an LM probability baseline. LM probability estimates are notoriously impacted by frequency effects [3], and so, if frequency effects were the source of classification performance, we would anticipate this baseline to match our modal difference vector classification performance.
>
>
>
> Q1: To clarify, the “Random” baseline is not a “chance” baseline. Figure 6 is simply showing that the modal difference vectors don’t provide extra predictive power over an arbitrary direction in vector space, for models <2B and for comparisons with the inconceivable modal category. In other words, the modal vectors found using the Hu et al. 2025b dataset do not generalize well. My explanation for this is that the model does not derive one consistent direction corresponding to “selectional restriction violations” at this scale. As described above, to achieve good performance on e.g., the Kauf dataset using vectors from the Hu et al. 2025b, the model must internally represent very different selectional restriction violations (on concreteness vs. animacy) consistently within its activations. If these two violations are represented differently, then there is no reason that the modal difference vector would outperform an arbitrary vector.
>
> I will include a discussion of this finding in Appendix Section B in the final submission.
>
> [1] https://alignment.anthropic.com/2025/modifying-beliefs-via-sdf/
> [2] Hu et al. 2025 Shades of Zero: Distinguishing Impossibility from Inconceivability
> [3] McCoy et al. 2024 Embers of Autoregression: Understanding Large Language Models Through the Problem They are Trained to Solve

---

### Official Review · Reviewer_8BZg · 2025-11-01

**Soundness:** 3
**Presentation:** 3
**Contribution:** 3
**Rating:** 6
**Confidence:** 4

**Summary:**

This paper investigates whether large language models encode internal representations of modal categories (probable, improbable, impossible, and inconceivable) and whether these internal representations align with human judgments.
Using Contrastive Activation Addition (CAA), the authors construct modal difference vectors, representing linear directions in hidden-state space that capture distinctions between modal categories.

**Strengths:**

- It is very interesting that these internal representations develop systematically with model scale, depth, and training progression, mirroring human developmental trajectories.
- The work builds on linguistic, philosophical, and cognitive frameworks of modality and imaginability, connecting results to human developmental and philosophical theories
- The correspondence between model-internal representations and human categorization behavior is compelling and well-analyzed.

**Weaknesses:**

- The study primarily focuses on short, declarative English sentences describing simple physical or conceptual scenarios. As a result, the findings may not generalize to contextualized modality. Expanding the dataset to include diverse syntactic and pragmatic expressions of modality would provide stronger evidence that the observed internal representations are not tied to a narrow stimulus set.
- Lack of analysis on confidence and uncertainty alignment. While the paper examines model–human alignment in categorical judgments, it does not analyze how models express confidence in those judgments and contrast between the narrative confidence (what the model says) and the representational confidence, what it encodes internally . It would be valuable to explore cases where the model is confidently wrong or hesitantly correct, and to compare the model’s internal certainty with the explicit “confidence” conveyed in its output probabilities.
- Some datasets (e.g., Hu et al. 2025a) are small; robustness of correlations might depend on a few stimuli.
- The reported findings are based on a small family of transformer models. It remains unclear whether the same modal geometry emerges across architectures or initialization seeds.

**Questions:**

- The logistic regression uses three modal difference vectors as features. How sensitive are the results to this choice?
- The authors employ correlation, MSE, and entropy metrics to compare model predictions and human judgments. Whether these metrics capture qualitative alignment as well as quantitative similarity, like category boundaries? For instance, do models misclassify in similar ways as humans do?

---

> ### Author Response · Authors · 2025-11-20
>
> Thank you for your detailed review and helpful comments! Find responses to your specific points below:
>
> W1: As described in Section 3.1, the datasets we employed vary in a number of ways, most saliently in the form of modal violation (e.g., concreteness violations, impossibility due to biology vs. physics, etc.). However, the datasets also do vary syntactically, with the Goulding dataset consisting of sentences in a future tense (e.g., “Someone is about to…”), and the remaining classification sentences being in the past tense. Additionally, Section 6 leverages the Tuckute et al. dataset, which consists of a very diverse corpus of sentences. When using this dataset, we find that our modal difference vector for impossible-inconceivable correlates well with imageability ratings on these diverse sentences, concordant with existing literature.
>
> Nonetheless, your point is well taken that the sentences are largely short, and fairly decontextualized. We believe that expanding our analysis to more pragmatically-complex scenarios is a promising area of further research, but properly doing so would require tackling issues such as figurative language, which are out of scope for the present article.
>
> W2. Interesting point! If I understand correctly, to do so exactly in the manner that you describe would require eliciting explicit modality classifications from the model, rather than overall probabilities of sentences. Following prior work, we elected to do the latter, as it is known that models underperform when faced with metalinguistic versions of tasks [1]. However, as you noted, we do compare against what we view as the gold-standard uncertainty metric for stimuli – human classification uncertainty. Here we find that a regression trained on modal difference vectors captures the classification uncertainty of human participants better than a regression trained using a probability feature.
>
> W3: This point is well taken, and it is for this reason that we separate out results by dataset in Figure 4!
>
> W4: While it is true that our results are limited to the standard transformer architecture, we analyze 11 models at various scales and across various model families. Across this variation, we mainly see a divergence in modal geometry as a result of scale, with <2B models performing worse than >=2B models. Larger models (across families), perform rather similarly (as evidenced by Figure 3(a)).
>
> Q1: Good question! In the final submission of the manuscript, we will include an ablation where we train regression models using subsets of the modal difference vectors. This will be found in the appendix.
>
> Q2: Glad you asked! See Table 4 in the appendix, where we include qualitative examples from the Goulding dataset (which contains human judgments of Possibility vs. Impossibility). In this table, we see a concrete example of the modal difference vector regression capturing human uncertainty (and human certainty!) better than the probability-based regression. We will include this table in the main text upon submission.
>
> Additionally, the results from Figure 4 are all about correlation to human subject response distributions (not experiments annotations). Even moreso than experimenter annotations, we view these judgment distributions as the gold standard modality labels. One can see by the fairly low MSE of the modal difference regressions that this feature space captures human judgments well.
>
> [1] Hu & Frank 2024 Auxiliary Task Demands Mask the Capabilities of Smaller Language Models

---

### Meta-Review · Area_Chair_jEfY · 2025-12-10

**Summary:**

This paper received consistent positive scores 6, 6, 6, 6. The main concerns raised include the limited diversity and scale of the dataset, the narrow range of evaluated models, the imbalance of dataset sentence distributions, the absence of strong baselines, and inconsistencies between the claims and the experimental findings. In the rebuttal, the authors responded to these concerns, and one reviewer indicated that the concerns about dataset scale and the inconsistencies were addressed. For the remaining concerns, the authors also provided detailed explanations. Therefore, the AC has decided to accept this paper.

**Reviewer Concerns:**

In the rebuttal, the authors explained that the datasets they used differ in several aspects and that one of the datasets contains a highly diverse corpus of sentences. They also clarified that their analysis covers multiple scales and spans various model families. One reviewer indicated that the concerns regarding dataset scale and the inconsistencies between the claims and the experimental findings were addressed. Therefore, I believe the concerns about the limited diversity and scale of the dataset, the range of evaluated models, and the inconsistencies have been addressed. However, since no new experimental results were provided in the rebuttal, the concern regarding the absence of strong baselines remains insufficiently addressed.

**Reviewer Scores:**

For reviewer bMoA, who indicated that most of his concerns were addressed, I believe he would be inclined to raise his score. For the other three reviewers, since the authors provided reasonable explanations, I think they would either maintain their current ratings or possibly raise them if they had been able to participate fully in the discussion.

---

### Decision · Program_Chairs · 2026-01-26

Accept (Poster)